# Neuronal Guidance Molecules in Bone Remodeling and Orthodontic Tooth Movement

**DOI:** 10.3390/ijms231710077

**Published:** 2022-09-03

**Authors:** Sinan Şen, Ralf Erber

**Affiliations:** 1Department of Orthodontics, University Medical Center Schleswig-Holstein, Campus Kiel, Christian Albrechts University, 24105 Kiel, Germany; 2Department of Orthodontics and Dentofacial Orthopedics, University of Heidelberg, Im Neuenheimer Feld 400, 69120 Heidelberg, Germany

**Keywords:** orthodontic tooth movement, bone remodeling, neuronal guidance molecules, Eph receptors, ephrins, Semaphorins

## Abstract

During orthodontic tooth movement, mechanically induced remodeling occurs in the alveolar bone due to the action of orthodontic forces. The number of factors identified to be involved in mechanically induced bone remodeling is growing steadily. With the uncovering of the functions of neuronal guidance molecules (NGMs) for skeletal development as well as for bone homeostasis, NGMs are now also among the potentially significant factors for the regulation of bone remodeling during orthodontic tooth movement. This narrative review attempts to summarize the functions of NGMs in bone homeostasis and provides insight into the currently sparse literature on the functions of these molecules during orthodontic tooth movement. Presently, four families of NGMs are known: Netrins, Slits, Semaphorins, ephrins and Eph receptors. A search of electronic databases revealed roles in bone homeostasis for representatives from all four NGM families. Functions during orthodontic tooth movement, however, were only identified for Semaphorins, ephrins and Eph receptors. For these, crucial prerequisites for participation in the regulation of orthodontically induced bone remodeling, such as expression in cells of the periodontal ligament and in the alveolar bone, as well as mechanical inducibility, were shown, which suggests that the importance of NGMs in orthodontic tooth movement may be underappreciated to date and further research might be warranted.

## 1. Introduction

Orthodontic tooth movement relies on bone resorption on the side where the periodontal ligament is compressed and bone augmentation on the side where the PDL is stretched. Ultimately, this allows a tooth to migrate through the alveolar bone. The elucidation of the biology of orthodontic tooth movement for the correction of dental malocclusions has gained importance in recent years [1,2,3,4].

During orthodontic tooth movement, mechanically induced remodeling occurs in the alveolar bone due to the application of orthodontic forces. Although bone cells of any anatomical localization (including osteoblasts, osteoclasts, as well as osteocytes) are sensitive to mechanical loads and thus adapt to changing external conditions [5,6], it is now recognized that force application to the periodontal ligament (PDL) and the fibroblasts localized there is crucial for orthodontic tooth movement. However, the exact mode of force absorption and subsequent mechanotransduction have not been fully elucidated nor have all the molecular factors involved in this process been identified and functionally characterized. The number of signaling molecules associated with orthodontic tooth movement to date exceeds 100 and ranges from pro-inflammatory cytokines and “classical” coupling factors such as RANKL (Receptor Activator of NF-κB Ligand) and OPG (Osteoprotegerin) to autophagy regulators and hypoxia mediators [7]. What has actually gained increasing attention recently with regard to the regulation of bone remodeling during orthodontic tooth movement is the putative involvement of neuronal guidance molecules. This seems, at first sight, somewhat surprising as these molecules were originally identified for their functions in neuronal development, modulation of cell positioning and tissue and organ patterning [8], but a closer look shows that the importance of these molecules in the regulation of bone homeostasis and orthodontic tooth movement may still be underestimated. Indeed, NGM-mediated interactions between the nervous system and the skeleton have been shown in different contexts [9] and direct NGM effects on bone homeostasis [10] as well as their involvement in tooth development were shown [11]. Intriguingly, numerous NGMs are still expressed in adult dental tissue [12,13,14] and their function here is only beginning to be understood. Thus, the aim of this narrative review is to summarize the known functions of NGMs in bone remodeling and bone homeostasis and to review the available literature on NGM functions in orthodontic tooth movement. Therefore, we will first briefly summarize the main processes and the known molecular factors involved in physiological bone remodeling. Then, separately for each family, we will introduce the NGMs for which a role in bone remodeling has been described and explain their functions. Finally, we will discuss the NGMs that have previously been associated with oral tooth movement and explain their potential importance in this process.

We should like to point out that this review is narrative in nature. In order to reduce the possible bias, a literature search including all four known families of NGMs was conducted. Databases consulted in the project included: PubMed, EBSCO, Web of Science and Scopus. The following queries were used for the search: “(bone remodeling OR bone homeostasis OR bone) AND “(axon guidance molecules OR neuronal guidance molecules OR axon path-finding OR axon guidance)”; (Orthodontic tooth movement AND (netrin* OR slit* OR semaphorin* OR ephrin* OR eph* OR eph-receptor*)”. We did not perform a specific quality assessment, which would hardly have been possible due to the basic research nature of most of the studies.

## 2. Bone Remodeling

As already mentioned, the prerequisite for orthodontic tooth movement is bone remodeling. This process will, therefore, first be briefly summarized in order to explain the possible functions of the NGMs in this context later. Apart from mechanically induced changes, bone, as a highly dynamic tissue, is able to maintain its structural integrity through constant remodeling. This occurs through the coordinated activity of bone-degrading osteoclasts, bone-forming osteoblasts, and their precursors. At the beginning of bone remodeling, hematopoietic progenitor cells are recruited and differentiate into osteoclasts. As the remodeling process continues, osteoclasts die by apoptosis, thereby allowing bone formation by newly recruited osteoblasts. Finally, osteoblasts differentiate and complete bone formation with mineralization of the newly formed bone matrix. These precisely controlled processes are governed by various diffusible local and systemic factors that are both released from the bone matrix and secreted by osteoblasts and osteoclasts during the resorption phase [15]. In addition, direct cell–cell interactions between osteoblasts and osteoclasts also occur via membrane-bound ligands and receptors [16].

For the differentiation of osteoclasts from hematopoietic progenitor cells, M-CSF (macrophage-colony stimulating factor) and RANKL (receptor activator of nuclear factor kappa β ligand) are crucial. M-CSF is produced by osteoblasts and cells of the stroma and mediates proliferation of hematopoietic osteoclast progenitor cells and increased expression of the RANKL receptor RANK. Binding of RANKL to RANK induces differentiation into bone resorbing osteoclasts. In this process, a number of signaling pathways are activated, which, among others, cumulate in the activation of the transcription factor NFATc1 (Nuclear Factor of Activated T-Cells, Cytoplasmic, Calcineurin-Dependent 1), DAP12 (DNAX activation protein of 12 kDa) and its co-receptors TREM-2 (Triggering receptor expressed on myeloid cells 2) and OSCAR (Osteoclast-associated receptor) which are necessary for its calcium-dependent activation [17].

This resorption process is counter-regulated by osteoprotegerin (OPG). OPG is a soluble mock receptor for RANKL and thus can compete its binding to RANK [18]. A variety of signaling pathways, such as the BMP-(bone morphogenic proteins) dependent or the canonical Wnt signaling pathways, subsequently induce the expression of the key transcription factor for osteogenic differentiation RUNX2 (Runt related transcription factor 2) in osteoblasts, under the control of which numerous osteogenic differentiation markers, such as ALP (Alkaline phosphatase), SPP1 (Osteopontin) or BGLAP (Osteocalcin) are activated [19].

Since the initiation of the necessary bone remodeling is dependent on mechanical forces (orthodontic forces), the conversion of mechanical loads into biological responses is critical for the initiation and progression of tooth movement. Over the years, a variety of molecules have been associated with bone remodeling during orthodontic tooth movement. Most commonly, by their very nature, factors that have already been linked to bone remodeling during embryonic development and to lifelong bone remodeling have been studied. A recent review [7] lists a total of 139 proteins of which, however, only Interleukin (IL)-1β, Cyclooxygenase (COX)-2 and Prostaglandin (PG)-E2, the osteoblast markers Osteocalcin and Runt-related transcription factor (RUNX)2, Receptor activator of nuclear factor kappa-B ligand (RANKL), Osteoprotegerin (OPG) and extracellular signal-regulated kinases (ERK)1/2 were investigated in 10 or more studies.

Only recently NGMs were identified to function in bone remodeling. To date, four conserved families of NGMs are identified in vertebrates: Netrins, Slits, Semaphorins and ephrins. Whereas Netrins, Slits and some Semaphorins are secreted molecules; ephrins and other Semaphorins are membrane bound. The respective transmembrane receptors have also been identified: UNC-40 (or DCC), Neogenin and UNC-5 family members are receptors for Netrins [20,21,22], Roundabout (ROBO) are receptors for Slit proteins [23], Neuropilin and Plexin are receptors for Semaphorins [24] and Eph receptors bind ephrins [25,26]. Whereas Netrins and Slits show rather selective binding, Semaphorins and ephrins are considerably promiscuous in receptor binding [24,25,26].

### 2.1. Neuronal Guidance Molecules in Bone Remodeling

#### 2.1.1. Netrins

The netrin family consists of four secreted members (Netrin 1–4) and the two transmembrane-bound members Netrin-G1 and Netrin-G2 [27].

Netrins have an important role in vascular development. For instance, the UNC5B receptor (uncoordinated 5b) is mainly vasculature specific and is expressed only in small neuronal subpopulations in the retina, ear and cerebellum and in very few cells outside the nervous system [28]. Netrin1 inhibits the migration of monocytes, lymphocytes and macrophages via activation of the UNC5B receptor. Furthermore, it is expressed by cells of the vascular endothelium, where it prevents cell migration into surrounding tissues. In sepsis or inflammatory conditions, Netrin1 is reduced and might thereby enhance leukocyte recruitment and migration [29]. Netrin1 is also involved in the regulation of cell motility. Via UNC5B and DCC (Deleted in Colorectal Cancer), Netrin1 regulates the small GTPases Rac and Rho, resulting in a change in the actin cytoskeleton [30]. Likewise, Netrin1 activates the MAP kinase (mitogen activated protein kinase) signaling pathway, interfering with a crucial step in the regulation of the actin cytoskeleton that controls the dynamics of cell motility [31]. In addition, Netrin1 can alter intracellular Ca2+ and cAMP levels as well as ion channel permeability [32].

Murine MC3T3-E1 pre-osteoblastic cells were shown to express the Netrin receptors Unc5b and Neogenin. During osteogenic differentiation induced by mineralization medium, Netrin-1 expression decreased, and exogenous Netrin-1 suppressed osteogenic differentiation. Furthermore, induction of osteogenic differentiation induced by BMP-4 was also suppressed by Netrin-1. Netrin-1 binding to Unc5b promoted osteoclast differentiation in vitro that could be prevented by antibody-mediated blockade of Netrin-1 or Unc5b. These in vitro findings were confirmed in Netrin-1-deficient (Ntn1-/-) mice [33]. A crosstalk between the nervous and skeletal systems mediated by Netrin-1 interactions with its receptor, DCC, was proposed from data derived from an osteoarthritis mouse model. Here, Netrin-1 secreted by osteoclasts was involved in sensory innervation of subchondral bone [34]. Bone remodeling, like skeletal embryonic development, is closely linked to angiogenesis. Indeed, roles in angiogenesis and neo-angiogenesis have also been described for various NGMs [35]. More recent data show that, also, Netrin-4 may link angiogenesis and bone remodeling. Endothelial Netrin 4 inhibits osteoclast differentiation in vitro and showed osteoprotective effects in an osteoporotic mouse model [36], but the effect of Netrin-4 does not seem to be limited to osteoclasts, as Netrin-4-dependent differentiation and migration of osteoblasts were also shown [37].

Tooth development is a complex process that requires the mutual interaction of embryonic oral epithelial cells and ectomesenchymal cells from the neural crest [38]. Netrin 1 and Netrin-3 are expressed in the developing tooth germ and might be involved in axon growth during tooth development. Netrin-1-deficient mice, however, did not show axon growth or tooth innervation alterations.

#### 2.1.2. Ephrins and Eph Receptors

The Eph receptors form the largest subgroup of receptor tyrosine kinases known to date. The human genome encodes a total of 14 members [39]. The Eph receptors are divided into two classes based on sequence homologies and their binding properties: nine EphA-receptors (EphA1-EphA8, EphA10) and five EphB-receptors (EphB1-EphB4, EphB6) [26,40,41]. The N-terminal, extracellular portion of membrane-bound Eph receptors includes the globular ephrin ligand-binding domain, a cysteine-rich region, and two fibronectin type III (FNIII) domains. The simple transmembrane domain is followed, intracellularly, by the juxtamembrane region, the tyrosine kinase domain, a “sterile-α-motif” (SAM) domain that regulates receptor oligomerization, and the C-terminal PDZ-binding domain (PDB) that mediates signal transduction [42].

Eph-receptor-interacting ephrin ligands were not characterized until several years after the discovery of Eph receptors. The first ephrin ligand-A1, as well as its corresponding receptor EphA1, was identified in a hepatocellular carcinoma [43].

In the human genome, eight ephrin ligands are known to date, which are also divided into two classes based on their different receptor binding properties and their different anchorage in the cell membrane: Ephrin-A ligands (EphA1-EphA5) and Ephrin-B ligands (EphrinB1-EphrinB3). Ephrin-A ligands are connected to the cell membrane via glycosylphosphatidylinositol anchors. In contrast to Ephrin A ligands, Ephrin-B ligands anchor via a simple transmembrane domain. Ephrin-B ligands have five–six tyrosine residues on the intracellular side that can be phosphorylated and a C-terminal PDZ-binding domain [39,44]. The EphA receptors bind preferentially to ephrin-A ligands, whereas the EphB receptors bind preferentially to ephrin-B ligands [45]. Within classes, binding specificity is low, suggesting high redundancy in the system.

The exceptions are the EphA4 and EphB2 receptors, which can bind other ephrin-B ligands as well as ephrin-A5. Also an exception is the EphB4 receptor, which binds only the ephrin-B2 ligand [44,46,47].

A special feature of the Eph–ephrin interaction is the possibility of bidirectional signal transduction [45]. If the direction of signal transduction is from the ligand to the receptor or dependent on receptor activation, then this is referred to as forward signaling. The opposite direction of signal transduction is referred to as “reverse signaling”.

Via such bidirectional signal transduction, members of the Eph/Ephrin family could thus be able to simultaneously regulate osteoclast and osteoblast activity. Indeed, Zhao and colleagues [48] demonstrated that ephrin-B2–EphB4-dependent signal transduction is involved in the reciprocal control of osteoclasts and osteoblasts: EphB4-dependent ephrin-B2 reverse signaling in osteoclasts inhibits their differentiation, and at the same time, activation of EphB4-receptor-dependent forward signaling in osteoblasts by ephrin-B2 stimulates osteoblast activity and differentiation. In sum, this led to bone formation.

Ephrin-B2 was upregulated during osteoclast differentiation in a NFATc1 (nuclear factor of activated T-cells, cytoplasmic 1)- and c-Fos-dependent manner, which suggested a possible involvement of ephrin-B2-dependent signaling in osteoclast differentiation. Stimulation of ephrin-B2 ligands on osteoclasts by EphB4 (soluble oligomeric receptor chimeras) inhibited osteoclastogenic differentiation. The same effect was observed by overexpressing ephrin-B2 in osteoclasts. In both cases, inhibition of osteoclastogenic differentiation was independent from RANKL or M-CSF, whose expression was unaffected by stimulation with EphB4. Conversely, loss of ephrin-B2 expression in osteoclasts precursors, through the use of “small interfering” (si-) RNAs directed against ephrin-B2, caused an enhancement of osteoclastogenic differentiation and activity. Similar results were obtained when macrophages from conditional ephrin-B2 knock-out mice were studied.

It was further found that one of the downstream signal transduction mechanisms that contributed to inhibition of osteoclast differentiation after activation of ephrin-B2 reverse signaling involves inhibition of the expression of Fos and NFATc1. Overexpression of endogenous Fos and/or NFATc1 could revert this inhibitory effect. The effects of EphB4-receptor-activation-dependent forward signaling were also investigated. Activation of the EphB4 receptor on osteoblasts by ephrin-B2 ligands (soluble oligomeric ligand chimeras), resulted in enhanced osteogenic differentiation, characterized by induction of expression of alkaline phosphatase, Collagen 1a1 (Col1a1), Osteocalcin (Bglap), Osterix (SP7), and Runt related transcription -factor 2 (Runx2), the major transcription factor in osteoblastogenic differentiation. These effects could be reversed by siRNA directed against EphB4.

Activation of EphB4 in osteoblasts activates the MAP kinases ERK1/2, while simultaneously inhibiting the Rho GTPase RhoA. Thus, activation of EphB4 on osteoblasts involves a signal transduction pathway involving activation of ERK1/2 and inhibition of RhoA, which ultimately stimulates osteoblastogenic differentiation. More recently, EphB4 signaling was also shown to be required during bone fracture repair, suggesting a role for Eph–ephrin signaling not only during life-long remodeling but also in trauma [49]. Finally, Zhao and colleagues demonstrated that overexpression of EphB4 in bone led to an increase in bone mass and bone density in transgenic mice with the EphB4 gene under the control of the Collagen 1a1 promoter.

A few years after the identification of EphB4–ephrin-B2 functions in bone remodeling, the same lab showed that bidirectional activation of the ephrin-A2–EphA2 signaling pathway triggers the opposite of what had been observed in osteoblasts and osteoclasts for ephrin-B2 and EphB4. Activation of EphA2 receptor forward signaling in osteoblasts by ephrin-A2 expressed by osteoclasts led to inhibition of osteoblastic differentiation in osteoblasts. At the same time, ephrin-A2-dependent reverse signaling in osteoclasts induces their differentiation and activity. Taken together, EphA2–ephrin-A2 signal transduction in osteoblasts and osteoclasts leads to the enhancement of osteoclastogenesis and inhibits osteoblastogenesis, this, in sum, caused bone resorption [50]. To further investigate the function of ephrin-A2 in osteoclasts, Irie and colleagues [50] transduced osteoclast progenitor cells with a retroviral vector encoding ephrin-A2. The corresponding osteoclasts showed clear signs of differentiation and were able to degrade bone mass in vitro. EphA2-expressing osteoblasts were able to induce differentiation of ephrin-A2-overexpressing osteoclasts. This induction was abolished as soon as osteoblasts lacking the EphA2 receptor were co-cultured. Interestingly, it was further shown that activation also of the EphA2 receptor on osteoclasts led to activation of osteoclastogenic differentiation. Furthermore, it could be shown that EphA2 receptor activation was at least partly dependent on ephrin-A2, which is detached from the osteoclasts themselves by the activity of certain matrix metalloproteinases (MMPs). Thus, in osteoclasts, both activation of the reverse signaling pathway via ephrin-A2 and the classical EphA2-receptor-dependent forward signaling pathway induced osteoclast differentiation. Finally, the authors provided indirect evidence for a negative effect of EphA2 receptor activation on osteoblast differentiation. Osteoblasts isolated from EphA2-deficient newborn mice showed accelerated and more pronounced differentiation under osteoblastogenic culture conditions compared with wild-type osteoblasts.

In the interactions of Ephs and ephrins described above direct cell heterotypic interactions between osteogenic and osteoclastic cells are necessary, whether these exist in vivo has been questioned. At least in mice, via an elegant approach involving two-photon intravital microscopy, the laboratory of Masaru Ishii has proven that osteoblasts and osteoclasts are regulated via direct cell–cell contact between these two cell types [51]. Thus, coupling of the activity of osteoblasts and osteoclasts via Ephrin–ephrin interactions remains possible and underlines the potential importance of these NGMs for bone remodeling.

This is further illustrated in findings supporting the importance of other Ephs and ephrins in the context of skeletal development and homeostasis: Ephrin-A2, EphA4, EphA7 and ephrin-A5 are involved in the early stages of avian skeletal development [52]. Frontonasal dysplasia and coronal craniosynostosis in humans are cranial defects caused by loss-of-function mutations in the EFNB1 gene coding for ephrin-B1 [53]. Mice deficient for ephrin-B1 have a severe skeletal phenotype with reduced bone formation [54]. The targeted deletion of ephrin-B1 in under the control of the Osterix promoter caused developmental growth plate defects [55] whereas overexpression of ephrin-B1 in osteoblasts enhanced bone formation [56]. The loss of ephrin-B2 caused an increase in trabecular bone volume suggesting a role for ephrin-B2 in endochondral ossification [57]. A central role for Eph–ephrin signaling in bone homeostasis is further supported by the work of Allan and colleagues [58], who demonstrated parathyroid hormone (PTH)- and parathyroid hormone 1 receptor (PTHR1)-dependent induction of ephrin-b2 in osteoblasts and that ephrin-B2 and Ephb4-dependent interactions within the osteoblast population induced osteogenic differentiation. Together with the work of Takyar and colleagues [59], this suggested that PTH-driven EphB4 activation in osteoblasts is necessary for at least the late stages of osteogenic differentiation. EphA4 deficient mice have reduced trabecular bone volume and overall bone loss due to greater bone resorption capacity of mature osteoclasts, which also implies a multifaceted role of EphA2-dependent signaling in osteoclasts and their precursors [50,60]. In osteoclast precursors, EphB2 and ephrin-B1 were shown to be negative regulators of osteoclast differentiation [61,62]. Together, these findings show that numerous Eph–ephrin interactions contribute to spatial and temporal osteoblast and osteoclast regulation contributing to bone remodeling. Within the context of orthodontic tooth movement, where bone remodeling is dependent on mechanical loads, it is noteworthy the EphB2 and ephrin-B1 were shown to be upregulated by mechanical load using a mouse tibia model [63].

During tooth development, shortly after gastrulation, neural crest cells from the posterior mesencephalon and anterior hindbrain migrate into the maxillary region and form the mesenchyme under the epithelium. This occurs under the influence and with the directive activity of various neuronal guidance molecules including ephrin ligands and Eph receptors such as ephrin-A2, ephrin-B1, ephrin-B2, EphA3, EphA4, EphA7, EphA9, EphB2, EphB3 and EphB4 [64]. In particular, the expression of the ephrins and Eph receptors ephrin-A1-A5, and EphA2, EphA3, EphA4, EphA7 and EphA8 in the trigeminal ganglion plays a major role in tooth development [14]. As development continues, the epithelium of the oral bay invaginates into the mesenchyme and forms the primary epithelial ligament. Two processes develop from this ligament, the lamina dentalis and the lamina vestibularis. Whereas the lamina vestibularis separates the lips or cheeks from the alveolar ridge, the teeth develop from the lamina dentalis. Initially, the outwardly directed lamina dentalis forms epithelial bulges surrounded by compacted mesenchyme. These epithelial bulges rapidly enlarge and enclose the underlying mesenchyme, forming a bell-shaped structure. At this stage, differentiation of odontoblasts and ameloblasts occurs, initiating the formation of the hard tissues, dentin and enamel. The importance of ephrins and Eph receptors in tooth development is also underscored by in vitro data: in primary dental pulp stem cells ephrin-B1–EphB2 interaction induced osteogenic and odontoblastogenic differentiation [65].

#### 2.1.3. Semaphorins

Semaphorins form a family of more than 20 glycoproteins that is divided into eight classes based on sequence and structural similarities. Class 1 and 2 Semaphorins are found exclusively in invertebrates, classes 3–7 are specific to vertebrates, and viral Semaphorins are grouped in class V. Within the classes, a letter coding designates the different semaphorin molecules. Most Semaphorins are membrane-associated, similar to ephrins. In contrast, classes 2, 3 and 5 of the Semaphorins are secreted, soluble proteins. The structure of Semaphorins is similar across classes and is characterized by the cysteine-rich plexin, semaphorin, and integrin (PSI) domain and the Sema- domain common to all [66,67].

Semaphorin-dependent signaling occurs through transmembrane plexin receptors. In vertebrates, nine plexins in four families are known (Plexin A1–A4, Plexin B1–3, Plexin C1 and Plexin D1). Plexins are the only transmembrane receptors that can directly interact with small intracellular GTPases of the Rho family, such as RhoD, Rnd1 or Rac1, via an intrinsic GTPase-activating (GAP) cytoplasmic domain [68]. Activation of GTPases after Plexin activation by Semaphorins occurs through GEFs (guanine nucleotide exchange factors) such as FARP2 (FERM, RhoGEF and Pleckstrin Domain Protein 2) that interact directly with Plexin domains after Semaphorin binding [69]. Rho family GTPases are closely associated with cell motility by controlling reorganization of the actin cytoskeleton following activation of the cell motility cycle, which is necessary for the execution of repulsive or attractive movements mediated by Semaphorins [70]. Although the Plexins have structural homologies with receptor tyrosine kinases, they do not possess tyrosine kinase activity themselves and the detailed process of signal transduction after activation is therefore not yet clear. Most Semaphorins bind plexin receptors directly. The secreted class 3 Semaphorins are an exception. These require an initial interaction with a Neuropilin (NRP) that acts as a co-receptor. The two known Neuropilins, NRP1 and NRP2, can also interact with other ligands and receptors and are thus involved in a variety of other signal transduction pathways [71]. However, they are not able to transduce signals alone [72]. 

Functions in the maintenance of bone homeostasis have been shown in recent years for several Semaphorins in interaction with their receptors and coreceptors. Since the role of Semaphorin 3A (Sema3A) and its receptors are crucial for orthodontic tooth movement, only their roles will be discussed in detail.

Already in the nineties of the last millennium, two knockout mouse models demonstrated the importance of Sema3A for normal bone development. Although in both cases exon 1 of the Semaphorin 3A gene was replaced, in one case by a neo cassette [73] and in the other case by a LacZ neo cassette [74], the phenotypes were not exactly identical. However, common to both was a skeletal phenotype that was further characterized by marked osteopenia in the case of the mouse model generated by Taniguchi and colleagues. A comparable phenotype was observed in mice in which the Neuropilin 1 gene was replaced by a mutant variant (Nrp1Sema-) that was no longer able to bind Sema3A [75]. This result indicated that the Neuropilin 1 receptor is required for Sema3A function during bone homeostasis. The latter two mouse models were recently used to study in detail the function of Sema3A during normal conchdral development and bone homeostasis [76]. In this study, Sema3A was shown to have a dual function in bone remodeling by controlling both osteoclast and osteoblast activity. Hematopoietic osteoclast progenitor cells co-cultured with osteoblasts differentiated into osteoclasts only when osteoblasts were Sema3A deficient. Exogenous Sema3A was able to prevent osteoclastogenic differentiation in bone marrow-derived macrophages, but only if stimulation with Sema3A occurred before the addition of RANKL. Moreover, the Sema3A effect was dependent on the expression of Nrp1 on osteoclasts. Interestingly, Sema3A itself was responsible for the induction of Nrp1; in contrast, RANKL was able to prevent this via an NFκB-dependent signaling pathway. Both suggest that Nrp1 is the crucial co-receptor for Sema3A-dependent regulation of osteoblasts and osteoclasts. However, actual signal transduction occurs via the Plexin A1 receptor (PlexnA1). Nrp1 competes with TREM-2 for Plxna1 binding, so high Nrp1 abundance, as caused by Sema3A, prevents TREM-2 activation and thereby NFATc1-dependent differentiation of progenitor cells into osteoclasts. In addition, Sema3A decreased M-CSF-dependent migration of macrophages from bone marrow by preventing M-CSF-dependent activation of Rho GTPase.

In Sema3A-deficient animals, formation and differentiation of osteoblasts are decreased. In vitro, PlexnA1 was shown to interact with FARP-2 and activate the Rac1 GTPase after Sema3A stimulation. This resulted in nuclear translocation of the Wnt-dependent transcriptional co-regulator β-catenin and induction of osteogenic differentiation when co-stimulated with Wnt3A. In contrast, Wnt3A alone was unable to induce nuclear translocation of β-catenin. A pivotal role for Sema3A in bone homeostasis is further emphasized by an estrogen-induced osteocyte expression of Sema3A and a possible association of decreased Sema3A serum levels with age [77].

However, controversy exists regarding the exact role of Sema3a in bone remodeling when interpreting data from mice with neuron-specific Sema-3a deficiency. In contrast to global Sema3A deficiency, mice with an osteocyte-specific Sema3A deficiency either under the regulation of the Osterix or the Col1 promoter (Osx-Cre;Sema3a^fl/fl^, Col1-Cre;Sema3a^fl/fl^) have normal bone mass, however neuron-specific Sema3A deficiency ((Syn1-Cre;Sema3a^fl/fl^ and Nestin-Cre Sema3a^fl/fl^) caused an osteopenic phenotype in mice that might be related to reduced sensory innervation of the trabecular bones in these animals [78].

In addition to Sema3A, other Semaphorins have been implicated in bone remodeling. Sema6D, via the Plexin-A1 receptor, promotes osteoclast differentiation, involving triggering receptor expressed on myeloid cells 2 (Trem-2) and adaptor molecule DNAX- activation protein 12 (DAP12) [79]. In contrast, Sema3E decreased osteoclast formation from macrophages [80]. Sema7A was shown to effect both osteoblasts and osteoclasts. Sema7A stimulates the motility of osteoblast precursors via MAPK activation and promotes mature osteoclast formation by enhancing fusion of osteoclast precursors [81]. 

Several Semaphorins are expressed during different phases of tooth development, for instance Semaphorins 3A, 3C, 3F, 4F, 5B, 6A, 6B and 6C were highly expressed in development and then decreased, Semaphorins 3B, 4A and 7A increased during development whereas Semaphorins 3E, 4B, 4C, 4D, 4G and 5A remained unregulated or showed no temporal or spatial patterning. Specific functions of Semaphorins during tooth development are mostly not fully clear. Sema3A was proposed to act as an essential chemorepellent that regulates incisor tooth germ innervation at early developmental stages [82] and Sema7A might be involved in the terminal innervation of the dentin–pulp complex [83].

#### 2.1.4. Slits

Members of the Slit family were first discovered in Drosophila and described as secretory proteins secreted by a small subpopulation of glial cells along the forming central nervous system. Specific to Slits is their binding to the Roundabout receptor (ROBO), a transmembrane protein found on axonal growth cones. Slits [84,85] prevent axons from crossing the midline [86]. Organisms with mutations within this Slit gene show a breakdown of the regular arrangement of longitudinal and commissural axons. The same has been shown for mutations within the ROBO gene. In vertebrates, three homologous variants of Slits are known, designated Slit1–3 [87]. The Slit proteins are characterized by four N-terminal leucine rich repeats (LRRs) designated D1 to D4. These are followed by six epidermal growth factor (EGF)-like domains, one laminin G-like domain, three additional EGF-like domains, and a C-terminal cysteine knot domain. In vertebrates four ROBO receptors exist (ROBO1–4). The ectodomains of ROBO1 to 3 are similar in structure and are reminiscent of cell adhesion molecules. Five immunoglobulin (Ig)-like domains, (Ig1–5), and three fibronectin-3 (FN3)-like domains are found. ROBO4 has only two Ig- and FN3-like domains each and, unlike ROBO1 to 3, it is expressed in the endothelium rather than the central nervous system. The part of ROBOs projecting into the cytosol is evolutionarily only weakly conserved. However, four strongly conserved motifs are found (CC0-CC3) that appear in specific combinations depending on the respective receptor. [87].

A crucial function of Slit–ROBO interactions is the control of cell motility, which is dependent on Slit Robo Rho GTPase activating protein1 (srGAP1) binding to the CC3 domain of (mammalian) ROBO1. It leads to Slit-dependent inactivation of the Rho GTPase Cell division cycle 42 (Cdc42), which has a direct effect on the actin cytoskeleton and simultaneously increases the activity of the Rho GTPase Rac1. This in turn strengthens cell–cell contacts mediated by cadherin [88,89] and prevents further motility. The adaptor protein Dreadlock (Dock/Nck) also binds CC3 and mediates SLIT-induced recruitment of Son of Sevenless (SOS) that, in turn, leads to activation of the Rho GTPase Ras, thus influencing the dynamic change of the cytoskeleton [90,91]. Slit2 and ROBO may also interact with other neuronal guidance molecules: ROBO affects the Netrin receptor DCC, leading to inactivation of this receptor. Slit2 enters into a complex with Netrin1 and thereby inhibits Netrin1/DCC interaction, inducing apoptosis [92,93]. Early evidence for the involvement of Slits in the development of mineralized tissue came from observations on chicken embryos. For instance, Slit2 was expressed in branchial arches and in long cartilage structures and cells of the periosteum [94]. More recently accumulating data suggested a role for Slit dependent signaling in bone metabolism. Whereas Slit1 is hardly found in bone tissue, Slit2 and Slit 3 are expressed in osteoblasts and osteoclasts, although controversy exists about Slit3 expression and functions [95,96]. The corresponding receptors, Robo1 and Robo3, are found in osteoclasts and Robo1 and Robo2 are expressed in osteoblasts. Although it is accepted that Slit2 via binding to the Robo1 receptor and Cdc42 inhibition attenuates osteoclast differentiation [97], a role for Slit2 during osteoblastogenesis remains controversial. Sun and colleagues [98] showed Slit2- and Robo-receptor-dependent inhibition of osteoblast differentiation, however, Park and colleagues [97] could not confirm any Slit2-mediated changes in osteoblast differentiation in primary mouse osteoblasts. Slit3’s function in bone remodeling is also controversial. Kim and colleagues [95] introduced Slit3 as a possible coupling factor of osteoclast and osteoblast activity during bone remodeling by demonstrating that osteoclast-derived Slit3 promoted bone formation by activating osteoblasts via a pathway involving β-catenin and concurrently attenuated bone resorption by inhibiting Rac1 in osteoclasts. Interestingly, osteoclast-specific deficiency of Slit3 caused an increase in bone resorption and decreased bone mass, whereas osteoblast-specific deletion of Slit3 had no effect on bone mass. However, Xu and colleagues [99] reported conflicting data: osteoblast-specific Slit3-deficient mice showed a decrease in bone mass and osteoclast-specific Slit3-deficient mice had no change in bone mass. In their model, Slit3 was strongly expressed in osteoblasts but hardly expressed in osteoclasts; a finding that was confirmed by the work of Li and colleagues [96]. Thus, the actual role of Slit3 in bone remodeling cannot be defined conclusively as yet.

The possible role of the Slits and Robo receptors for tooth development were characterized in a mouse model. Slit1 was specifically expressed in enamel nodes. Slit2 and 3 were expressed in the bone mesenchyme surrounding the tooth, which later develops into a periodontal structure. Furthermore, Robo1 and 2 were shown be expressed in the mesenchyme of the tooth and jaw during tooth development [13].

### 2.2. Neuronal Guidance Molecules in Orthodontic Tooth Movement

#### 2.2.1. Ephrins and Eph Receptors

As mentioned above, functions in the maintenance of bone homeostasis have been defined for certain ephrin and Eph receptors. Within this context it should be noted once again that a distinctive feature of ephrin ligands is that, like their receptors, they are also capable of transmitting signals.

Bidirectional signaling between ephrin-B2 on osteoclasts and its receptor EphB4 on osteoblasts suppressed osteoclast differentiation while stimulating osteogenic differentiation of osteoblasts [48]. Likewise, bidirectional signaling, in this case between ephrin-A2 and EphA2 produced opposite results in osteoclasts and osteoblasts [50]. Taken together, bidirectional activation of ephrin-B2 and EphB4 resulted in bone formation whereas activation of ephrin-A2 and EphA2 resulted in bone resorption. These data suggested a high importance for the ephrin–Eph system in bone homeostasis. Therefore, a possible role of these signaling pathways in bone remodeling during orthodontic tooth movement was investigated in our laboratory. As previously mentioned, fibroblasts of the PDL (PDLF) are the primary recipients of orthodontic forces. However, mechanically modulated expression changes of ephrins had been described only in arterial endothelial cells [100,101]. Data for PDLF were not available at that time. In our studies, primary human PDLF responded to experimental mechanical stretch with an induction of ephrin-B2 expression and a decrease in ephrin-A2 expression. Compression forces had an opposite effect [102,103]. Stretch induced ephrin-B2 expression via a signaling pathway involving FAK (Focal Adhesion Kinase), GTPase Ras, ERK1/2 (Extracellular signal Regulated Kinases) MAP (Mitogen-Activated Protein) kinases, and transcription factor SP1 (Specifity Protein 1) [102]. Ephrin-B2-dependent osteogenic differentiation, clearly demonstrated by the expression of osteogenic differentiation markers and specific staining of mineralized matrix in primary human osteoblasts of the alveolar ridge, could be attributed to a signaling pathway activated after receptor stimulation, which also involved Ras and ERK1/2 [102]. Compression forces caused induction of ephrin-A2 and attenuated ephrin-B2 in PDLF. In alveolar ridge osteoblasts, activation of the EphA2 receptor by exogenous ephirn-A2 resulted in suppression of Ras activation and phosphorylation of ERK1/2, thereby omitting activation of osteogenic differentiation markers such as RUNX2 or ALPL [104]. Different mechanical stimulations therefore led to selective induction of either ephrin-B2 or ephrin-A2, each with reciprocal consequences for osteoblasts of the alveolar ridge. Unlike the induction of ephrin-B2 after stretch, initially it remained unclear how ephrin-A2 is induced after mechanical stimulation with compression forces in PDLF. Using three independent PDLF populations, we showed that the induction of ephrin-A2 mRNA after compression is accompanied by, compared with activation after stretch, delayed phosphorylation and transcriptional activation of c-fos, which heterodimerizes with c-jun to the AP-1 (activator protein-1) transcription factor. Compression-dependent induction of c-fos and ephrin-A2 was also demonstrated at the protein level. To prove the involvement of both ERK1/2 and c-fos, both factors were selectively inhibited. Activation of ERK1/2 was prevented by the use of U0126, a specific inhibitor of ERK1/2 kinase MEK1 (MAP2K1), and inactivation of c-fos at the transcriptional level was achieved by using si (small-interference) RNA. Both strategies resulted in a significant decrease in ephrin-A2 expression, thus demonstrating the direct involvement of both GTPase Ras, ERK1/2 and c-fos in the compression-dependent induction of ephrin-A2 in PDLF (Figure 1) [103]. Osteogenic differentiation of subpopulations of PDLFs contribute to bone formation during tooth movement, thus loss of this potential is relevant to bone remodeling during tooth movement. In previous experiments, we observed a marked ephrin-A2-dependent decrease in the expression of osteogenic markers in both alveolar ridge osteoblasts and PDLFs. Interestingly, this was even more pronounced in PDLF than in osteoblasts. A possible cause of this enhancing effect could be a higher abundance of ephrin-A2 in the PDLF population. Indeed, PDLF showed a significant induction of ephrin-A2 after EphA2 activation by exogenous ephrin-A2. This was accompanied by rapid activation of ERK1/2 and induction of c-fos. This implies that ephrin-A2 may enhance its own expression in the PDLF population, thereby contributing to the marked effect of ephrin-A2 on the inhibition of osteogenic differentiation in PDLF [103].

Our in vitro studies [102,103,104] have recently been complemented by animal studies by other authors that also provided evidence for the involvement of the Ephrin–Eph pathway in the regulation of bone remodeling during orthodontic tooth movement. Hou and colleagues used a rat model of orthodontic tooth movement and demonstrated strong ephrin-B2 staining in compressed periodontal tissues including PDLF and osteoclasts. More insight into ephrin-B2-dependent osteoclast activation was gained using RAW264.7 murine macrophage cells were compression induced the expression of ephrin-B2 along with the osteoclast markers nuclear factor of activated T cells cytoplasmic 1 (NFATc1) and calcitonin receptor (CTR). Moreover, the authors confirmed compression-force-dependent decreases of ephrin-B2 and EphB4 in ST2 murine stromal cells that are able to differentiate into osteoblast-like cells in response to osteogenic stimuli in vitro [105]. Only very recently, Jiang et al. focused on investigating an ephrin–Eph role on the tension side of orthodontic tooth movement. Also using a rat model, it was shown that local administration of a selective small molecule EphB4 receptor inhibitor targeting the kinase domain, shifted bone remodeling activity toward bone resorption after application of orthodontic forces, because EphB4 forward signaling was blocked in osteoblasts resulting in attenuated osteoblastic differentiation, whereas the extracellular domain of the EphB4 receptor was still able to initiate Ephrin-B2 reverse signaling and, thus, osteoclastogenesis [106].

#### 2.2.2. Semaphorins

As with the ephrins and their Eph receptors a crucial requirement for Sema3a involvement in the control of orthodontic tooth movement is direct or indirect modulation by mechanical forces in cells of the periodontium: fibroblasts of the periodontal ligament (PDLF) and osteoblasts of the alveolar ridge (OB). For Semaphorins, a shear-stress-dependent, microRNA-mediated regulation of Sema6A and Sema6D was demonstrated for the regulation of endothelial–pericyte interactions [107]. However, data for periodontal cells were not available. This led us to test mechanical regulation of Sema3A expression and its receptors neuropilin 1 (NRP1) and plexin A1 (PLEXNA1) in PDLF and osteoblasts of the alveolar bone [108]. 

Indeed, in PDLF we detected altered expressions of Sema3A: Sema3a was significantly induced by stretch forces in a time-dependent manner, whereas compression forces led to a significant reduction of Sema3A. With the exception of NRP1, which showed a significant reduction by stretch forces at individual time points, Sema3A receptors were either not significantly altered or, in the case of NRP1, significantly reduced by compression forces at most study time points. Therefore, there was no evidence for compensatory counter-regulation of both Sema3A receptors by mechanically induced modulation of Sema3A expression. Moreover, stretch also led to rapid significant mRNA expression induction of Osterix, the maxima of which preceded those of Sema3A, suggesting an involvement of Osterix (SP7) in Sema3A mechanoregulation (Figure 2). Because osteoblasts are also mechanosensitive, we also tested osteoblasts of the alveolar bone for mechanical regulation of Sema3A, its receptors and Osterix. Interestingly, no significant changes in expression were observed in osteoblasts after force application. Based on these in vitro results, the central role of PDLF in the initiation and regulation of tooth movement via NGMs seemed to be supported. We could further show that exogenous Sema3A stimulated the expression of osteogenic marker genes beyond the effect of osteogenic medium. Interestingly, the effect of Sema3A was more pronounced on early to mid-late markers; however, the effect of Sema3a was barely detectable in the absence of osteogenic preconditioning. Sema3A also had an effect on RANKL/OPG signaling. Here, the RANKL/RANK axis was negatively affected, whereas OPG expression remained largely unregulated. Results from murine systems support the involvement of plexin-associated GTPases as well as the WNT/-catenin signaling pathway in the Sema3A-activated signal transduction mechanism. Consistently Sema3A resulted in activation of Rac1 GTPase in osteoblasts of the alveolar bone. Sema3A caused significant transcriptional activation of β-catenin and its nuclear translocation accompanied by transcriptional induction of Runx2 and Osterix (SP7) (Figure 3) [108]. 

In vivo data from a mouse model supported our in vitro data and also suggested a role for sema3a in bone remodeling during orthodontic tooth movement. Kamei and colleagues, using a mouse model of orthodontic tooth movement, found increased expression of Sema3A on the tension side and locally administered recombinant Sema3A increased bone formation on the tension side and reduced the number of active osteoclasts at the compression side. From in vitro data obtained from primary mouse maxillary osteoblasts an Interleukin-1β-dependent Sema3 activation was suggested, which contradicts the data of Zhang et al. [109] showing direct mechanical induction of Sema3a in osteoblastic MC3T3-E1 cells but is in line with our findings from human primary cell osteoblasts of the alveolar bone. Since pro-inflammatory cytokines are part of the cellular response to orthodontic forces, it cannot be excluded that synergistic effects may be necessary for the activation of Sema3A expression in osteoblasts. At the same time, the lack of mechanical induction of Sema3A in osteoblasts again emphasizes the central role of PDL fibroblasts in orthodontic tooth movement.

The role of Sema3A in the regulation of bone remodeling has been recognized and increasingly includes functions in the periodontium as, for instance, reduced expression of Sema3A and Nrp1 in apical periodontitis in which decreased activity of the Sema3A-dependent pathway may be involved in periapical bone resorption [110]. In the rat mandible a temporary loss of Sema3A has been linked with the migration and activation of osteoclast progenitor cells [110]. Sema3A-dependent Nrp1 activation was also shown to trigger odontoblast differentiation of dental pulp stem cells via Wnt/β-catenin signaling [111]. Increasingly, links to tooth development and thus to hard tissue formation are also being described for neuronal functions of Sema3A [11], opening perspectives for the regeneration of dental hard tissue in vitro (“bioengineered teeth”) [112].

## 3. Conclusions

Several neuronal guidance molecules have emerged as regulators of bone development and homeostasis either by direct interactions with bone cells or by functionally coupling the nervous system with the skeleton. During orthodontic tooth movement the conversion of mechanical load, caused by the orthodontic appliances, into biological responses is crucial. Over the years a plethora involved in orthodontic tooth movement has been identified [7] including some NGMs that continue to show promise for their role in bone remodeling during orthodontic tooth movement, although more studies will undoubtedly be needed to confirm their potential roles.

However, their potential role should not be underestimated, as they possess a number of properties that predestine them as regulatory molecules during orthodontic tooth movement: many NGMs are involved in bone development and homeostasis; some have been shown to be directly regulated by key factors of bone homeostasis; their functions include the control of innervation and angiogenesis, both necessary during bone remodeling; many are expressed in adult hard tissues; and some have been shown to be mechanosensitive in different cell types. All this suggests that the importance of NGMs in bone remodeling and thus orthodontic tooth movement may be underappreciated to date and further research is therefore absolutely warranted. Further elucidation of the mechanisms by which NGMs are involved in bone remodeling in general, and in orthodontic tooth movement in particular, could pave the way for translational development of new therapeutics for the treatment of skeletal diseases as well as open up possibilities to perform orthodontic tooth movement without side effects.

## Figures and Tables

**Figure 1 ijms-23-10077-f001:**
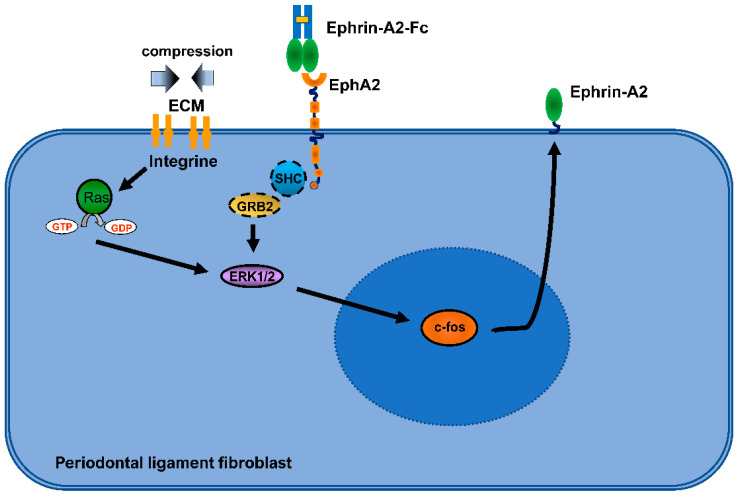
Putative signaling pathways involved in the regulation of ephrin-A2 in periodontal ligament fibroblasts (PDLFs). Both compression and stimulation by exogenous Ephrin-A2 induced ephrin-A2 expression via a signaling pathway involving GTPase Ras, ERK1/2 kinases and c-fos (ECM: extracellular matrix, SHC: Src homology 2 domain-containing adapter protein, GRB2: Growth factor receptor-bound protein 2).

**Figure 2 ijms-23-10077-f002:**
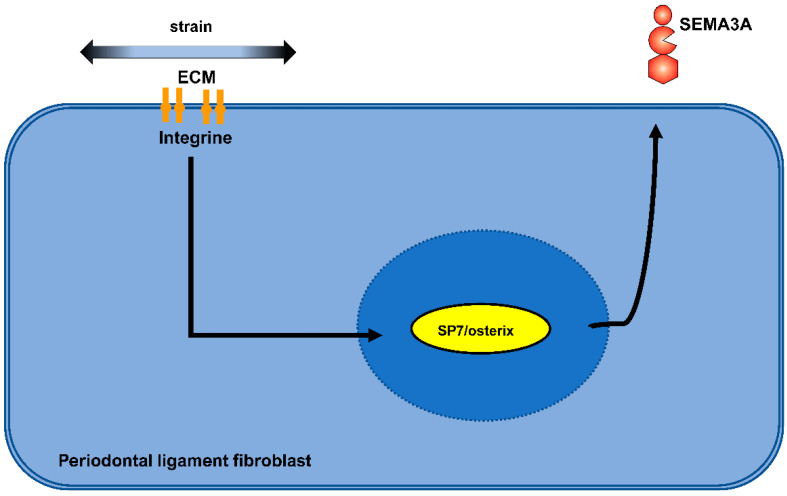
Sema3a was upregulated by stretch forces via Osterix induction in periodontal ligament fibroblasts.

**Figure 3 ijms-23-10077-f003:**
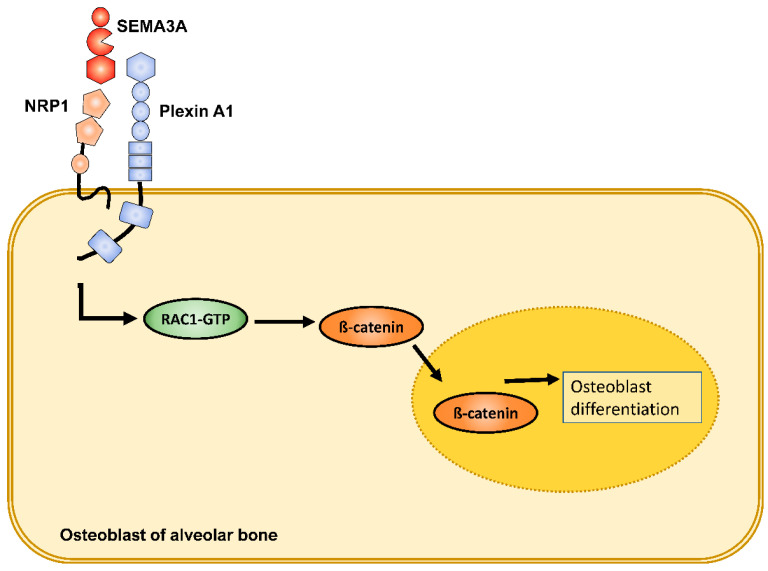
Exogenous Sema3A stimulation resulted in activation of Rac1 GTPase in osteoblasts of the alveolar bone. Sema3A caused transcriptional activation of β-catenin and its nuclear translocation accompanied by transcriptional induction of osteoblast differentiation via Runx2 and Osterix.

## Data Availability

The data presented in this study are available on request from the corresponding author.

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
