# Peer review of "Neuronal Guidance Molecules in Bone Remodeling and Orthodontic Tooth Movement"

_ijms, 2022, doi:10.3390/ijms231710077_

Round 1

Reviewer 1 Report

Dear Authors,

Orthodontic tooth movement relies bone resorption on the side where the periodontal ligament is compressed and bone augmentation on the side where the PDL is stretched. This narrative review summarizes the functions of neuronal guidance molecules in bone homeostasis and provides insight into the, admittedly scarce, literature on the functions of these molecules during orthodontic tooth movement.

I would like to congratulate with the authors for the interesting study they conducted.

The study was very interesting and it is of scientific interest and in line with the aims of the journal. 

However, there are some issues that should be addressed.

Abstract and References have to be modified according to the Instruction for Author.

However, there are some issues that should be addressed. 

Abstract

In my opinion the abstract is too short and it does not best describe the work the authors have done.

The abstract should be a total of about 200 words maximum. The abstract should be a single paragraph and should follow the style of structured abstracts, but without headings: 1) Background: Place the question addressed in a broad context and highlight the purpose of the study; 2) Methods: Describe briefly the main methods or treatments applied. Include any relevant preregistration numbers, and species and strains of any animals used. 3) Results: Summarize the article's main findings; and 4) Conclusion: Indicate the main conclusions or interpretations. The abstract should be an objective representation of the article: it must not contain results which are not presented and substantiated in the main text and should not exaggerate the main conclusions. (https://www.mdpi.com/journal/ijms/instructions).

Introduction

The Introduction Section not reports the rationale of the study and the aims.

The introduction should briefly place the study in a broad context and highlight why it is important. It should define the purpose of the work and its significance. The current state of the research field should be reviewed carefully and key publications cited. 

Please modify [1] [2] [3] [4] to [1-4]. Please in all the text.

Bone remodeling

-       “Before we go into more detail about the role of the NGMs during orthodontic tooth movement, we will first introduce the main representatives: In vertebrates, four conserved families of NGMs are identified to date: the Netrins, Slits, Semaphorins and ephrins”. Please rephrase, it is not clear.

Author Response

Manuscript ijms- 1869450 “Neuronal guidance molecules in orthodontic tooth movement”

We thank the reviewers for the close reading of our manuscript, for the interest in our work and for the valuable suggestions for improvement. The constructive suggestions from the reviewers led us to revise our manuscript accordingly. We believe these revisions affirm and extend the significance of our manuscript.

Response to Reviewer 1

Point 1: Abstract

In my opinion the abstract is too short and it does not best describe the work the authors have done.

The abstract should be a total of about 200 words maximum. The abstract should be a single paragraph and should follow the style of structured abstracts, but without headings: 1) Background: Place the question addressed in a broad context and highlight the purpose of the study; 2) Methods: Describe briefly the main methods or treatments applied. Include any relevant preregistration numbers, and species and strains of any animals used. 3) Results: Summarize the article's main findings; and 4) Conclusion: Indicate the main conclusions or interpretations. The abstract should be an objective representation of the article: it must not contain results which are not presented and substantiated in the main text and should not exaggerate the main conclusions. (https://www.mdpi.com/journal/ijms/instructions).

Response 1:

We apologize for not having provided a meaningful abstract. In the revised version, we have now considered the aspects raised by the reviewer: we have lengthened the abstract to 198 words and added the required structure (but without subheadings). We hope to have achieved an improvement with this.

Revised text:

During orthodontic tooth movement, mechanically induced remodeling occurs in the alveolar bone due to the action of orthodontic forces. The number of factors identified to be involved in mechanically induced bone remodeling is growing steadily. With the uncovering of functions of neuronal guidance molecules (NGMs) for skeletal development as well as for bone homeostasis, NGMs are now also among the potentially significant factors for the regulation of bone remodeling during orthodontic tooth movement. This narrative review attempts to summarize the functions of NGMs in bone homeostasis and provides insight into the currently sparse literature on the functions of these molecules during orthodontic tooth movement. Search of electronic databases revealed Netrins, Slits, Semaphorins, ephrins and Eph receptors as NGMs with roles in bone homeostasis. Functions during orthodontic tooth movement, however, were only identified for Semaphorins, ephrins and Eph receptors. Here, the data show that crucial prerequisites for participation in the regulation of orthodontically induced bone remodeling, such as expression in cells of the periodontal ligament and in the alveolar bone, as well as mechanical inducibility, are given which suggests that the importance of NGMs in orthodontic tooth movement may be underappreciated to date and further research might be warranted. 

Point 2: Introduction

The Introduction Section not reports the rationale of the study and the aims.

The introduction should briefly place the study in a broad context and highlight why it is important. It should define the purpose of the work and its significance. The current state of the research field should be reviewed carefully and key publications cited.

Please modify [1] [2] [3] [4] to [1-4]. Please in all the text.

Response 2:

We did underestimate the importance of an introduction section in a narrative review. We apologize for our mistake. To address the concerns of the reviewer we have modified the introduction section and have extended the introduction trying to comply with the requirements for an introduction section brought up by the reviewer. Most importantly w added rationale and aims of the review. In addition, we modified the citation style as required.

Revised text:

Orthodontic tooth movement relies bone resorption on the side where the periodontal ligament is compressed and bone augmentation on the side where the PDL is stretched. Ultimately, this allows a tooth to migrate through the alveolar bone. The elucidation of the biology of orthodontic tooth movement for the correction of dental malocclusions has gained importance in recent years [1-4].

During orthodontic tooth movement, mechanically induced remodeling occurs in the alveolar bone due to the application of orthodontic forces. Although bone cells of any anatomical localization (including osteoblasts, osteoclasts, as well as osteocytes) are them-selves sensitive to mechanical loads and thus adapt to changing external conditions [5, 6], it is now recognized that force application to the periodontal ligament (PDL) and the fibroblasts localized there is crucial for orthodontic tooth movement. However, the ex-act mode of force absorption and subsequent mechanotransduction have not been fully elucidated nor have all the molecular factors involved in this process been identified and functionally characterized. The number of signaling molecules associated with orthodontic tooth movement to date exceeds 100 and ranges from pro-inflammatory cytokines and "classical" coupling factors such as RANKL (Receptor Activator of NF-κB Ligand) and OPG (Osteoprotegerin) to autophagy regulators and hypoxia mediators [7]. What has actually gained increasing attention recently with regard to the regulation of bone remodeling during orthodontic tooth movement is the putative involvement of neuronal guidance molecules. This seems, at first sight, somewhat surprising as these molecules were originally identified for their functions in neuronal development, modulation of cell positioning and tissue and organ patterning [8], but a closer look shows that the importance of these molecules in the regulation of bone homeostasis and orthodontic tooth movement may still be underestimated. Indeed, NGMs mediated interactions between the nervous system and the skeleton have been shown in different contexts [9] and direct NGM effects on bone homeostasis [10] as well as their involvement in tooth development were shown [11]. Intriguingly, numerous NGMs are still expressed in adult dental tissue [12-14] and their function here is only beginning to be understood. Thus, the aim of this narrative re-view was to summarize known functions of NGMs in bone remodeling and bone homeo-stasis and to review the available literature on NGM functions in orthodontic tooth movement. Therefore, we will first briefly summarize the main processes and the known molecular factors involved in physiological bone remodeling. Then, separately for each family, we will introduce the NGMs for which a role in bone remodeling has been described and explain their functions. Finally, we will discuss the NGMs that have previously been associated with oral tooth movement and explain their potential importance in this process.

Point 3: Bone remodeling

-       “Before we go into more detail about the role of the NGMs during orthodontic tooth movement, we will first introduce the main representatives: In vertebrates, four conserved families of NGMs are identified to date: the Netrins, Slits, Semaphorins and ephrins”. Please rephrase, it is not clear.

Response 3:

We agree with the reviewer and have thus rephrased the section and believe that the meaning has become clearer now.

Revised text:

Only recently NGMs were identified to function in bone remodeling. To date, four conserved families of NGMs are identified in vertebrates:  Netrins, Slits, Semaphorins and ephrins. While Netrins, Slits and some Semaphorins are secreted molecules; ephrins and other Semaphorins are membrane bound. The respective transmembrane receptors have also been identified: UNC-40 (or DCC), Neogenin and UNC-5 family members are receptors for Netrins [20-22], Roundabout (ROBO) are receptors for Slit proteins [23], Neuropilin and Plexin are receptors for Semaphorins [24]and Eph receptors bind ephrins [25, 26] . While Netrins and Slits, show rather selective binding, Semaphorins and ephrins are considerably promiscuous in receptor binding [24-26].

Reviewer 2 Report

01

In the Introduction, please state what the aims of this review are.

02

Since this is a narrative review, elaborate at the end of the Introduction the rationale for the sub-topics (Bone remodeling, Neuronal guidance molecules in bone remodeling, and Neuronal guidance molecules in orthodontic tooth movement).

03

For the same reason stated in my previous comment (this is a narrative review), how do the authors know that neutrins, ephrins and eph-receptors, semaphorins, and slits would be the only neuronal guidance molecules in bone remodeling to discuss about?

The same for neuronal guidance molecules in orthodontic tooth movement.

04

There are some sentences in the text without reference to a previous study (or studies) in order to give evidence to their statements. Without references, these statements would be mere assumptions or allegations by the authors of the manuscript. Therefore, each of the following sentences need at least one reference to back up their statement:

“What has actually gained increasing attention recently with regard to the regulation of bone remodeling during orthodontic tooth movement is the putative involvement of neuronal guidance molecules. This seems, at first sight, somewhat surprising as these molecules were originally identified for their functions in neuronal development, modulation of cell positioning and tissue and organ patterning, but a closer look shows that the importance of these molecules in the regulation of bone homeostasis and orthodontic tooth movement may still be underestimated. Indeed, NGMs mediated interactions between the nervous system and the skeleton have been shown in different contexts and direct NGM effects on bone homeostasis as well as their involvement in tooth development were shown. Intriguingly, numerous NGMs are still expressed in adult dental tissue and their function here is only beginning to be understood.”

“While Netrins and Slits, show rather selective binding, Semaphorins and ephrins are considerably promiscuous.”

“Netrin 1 and Netrin-3 are expressed in the developing tooth germ and might be involved in axon growth during tooth development. Netrin-1 deficient mice, however, did not show axon growth or tooth innervation alterations.”

“The EphA-receptors bind preferentially to ephrin-A ligands, whereas the EphB-receptors bind preferentially to ephrin-B ligands (102). Within classes, binding specificity is low, suggesting high redundancy in the system.”

“Functions in the maintenance of bone homeostasis have been shown in recent years for several Semaphorins in interaction with their receptors and coreceptors. Since the role of Semaphorin 3A (Sema3A) and its receptors are crucial for orthodontic tooth movement, only their roles will be discussed in detail.”

“Stretch induced ephrin-B2 expression via a signaling pathway involving FAK (Focal Adhesion Kinase), GTPase Ras, ERK1/2 (Extracellular signal Regulated Kinases) MAP (Mitogen-Activated Protein) kinases, and transcription factor SP1 (Specifity Protein 1). Ephrin-B2-dependent osteogenic differentiation, clearly demonstrated by the expression of osteogenic differentiation markers and specific staining of mineralized matrix, in primary human osteoblasts of the alveolar ridge could be attributed to a signaling pathway activated after receptor stimulation, which also involved Ras and ERK1/2. Compression forces caused induction of ephrin-A2 and attenuated ephrin-B2 in PDLF. In alveolar ridge osteoblasts, activation of the EphA2 receptor by exogenous ephirn-A2 resulted in suppression of Ras activation and phosphorylation of ERK1/2, thereby omitting activation of osteogenic differentiation markers such as RUNX2 or ALPL. Different mechanical stimulations therefore led to selective induction of either ephrin-B2 or ephrin-A2, each with reciprocal consequences for osteoblasts of the alveolar ridge. Unlike the induction of ephrin-B2 after stretch, initially it remained unclear how ephrin-A2 is induced after mechanical stimulation with compression forces in PDLF. Using three independent PDLF populations, we showed that the induction of ephrin-A2 mRNA after compression is accompanied by, compared with activation after stretch, delayed phosphorylation and transcriptional activation of c-fos, which heterodimerizes with c-jun to the AP-1 (activator protein-1) transcription factor. Compression-dependent induction of c-fos and ephrin-A2 was also demonstrated at the protein level. To prove the involvement of both ERK1/2 and c-fos, both factors were selectively inhibited. Activation of ERK1/2 was prevented by the use of U0126 a specific inhibitor of ERK1/2 kinase MEK1 (MAP2K1), and inactivation of c-fos at the transcriptional level was achieved by using si (small-interference) RNA. Both strategies resulted in a significant decrease in ephrin-A2 expression, thus demonstrating the direct involvement of both GTPase Ras, ERK1/2, and c-fos in the compression-dependent induction of ephrin-A2 in PDLF (Figure 1). Osteogenic differentiation of subpopulations of PDLFs contribute to bone formation during tooth movement, thus loss of this potential is relevant to bone remodeling during tooth movement. In previous experiments, we observed a marked ephrin-A2-dependent decrease in the expression of osteogenic markers in both alveolar ridge osteoblasts and PDLFs. Interestingly, this was even more pronounced in PDLF than in osteoblasts. A possible cause of this enhancing effect could be a higher abundance of ephrin-A2 in the PDLF population. Indeed, PDLF showed a significant induction of ephrin-A2 after EphA2 activation by exogenous ephrin-A2. This was accompanied by rapid activation of ERK1/2 and induction of c-fos. This implies that ephrin-A2 may enhance its own expression in the PDLF population, thereby contributing to the marked effect of ephrin-A2 on the inhibition of osteogenic differentiation in PDLF.”

“is in line with our findings from human primary cells”

05

The authors might want to give a number to the following references in the text:

“Tooth development is a complex process that requires the mutual interaction of embryonic oral epithelial cells and ectomesenchymal cells from the neural crest (Jernvall and Thesleff, 2000).”

“To further investigate the function of ephrin-A2 in osteoclasts, Irie and colleagues (…)”

“EphA8 in the trigeminal ganglion plays a major role in tooth development (Luukko et al., 2005).”

“In primary dental pulp stem cells ephrin-B1-EphB2 interaction induced osteogenic and odontoblastogenic differentiation (Wang et al., 2013; Wang et al., 2014).”

“However, Xu and colleagues and Li and colleagues reported conflicting data”

Author Response

Manuscript ijms- 1869450 “Neuronal guidance molecules in orthodontic tooth movement”

We thank the reviewers for the close reading of our manuscript, for the interest in our work and for the valuable suggestions for improvement. The constructive suggestions from the reviewers led us to revise our manuscript accordingly. We believe these revisions affirm and extend the significance of our manuscript.

Response to Reviewer 2

Point 1:

In the Introduction, please state what the aims of this review are.

Point 2:

Since this is a narrative review, elaborate at the end of the Introduction the rationale for the sub-topics (Bone remodeling, Neuronal guidance molecules in bone remodeling, and Neuronal guidance molecules in orthodontic tooth movement).

Response 1 and 2:

Our response: We have followed the reviewer’s suggestion and stated the rationale for subtopics and aim of the review in the introduction section.

Revised text:

Orthodontic tooth movement relies bone resorption on the side where the periodontal ligament is compressed and bone augmentation on the side where the PDL is stretched. Ultimately, this allows a tooth to migrate through the alveolar bone. The elucidation of the biology of orthodontic tooth movement for the correction of dental malocclusions has gained importance in recent years [1-4].

During orthodontic tooth movement, mechanically induced remodeling occurs in the alveolar bone due to the application of orthodontic forces. Although bone cells of any anatomical localization (including osteoblasts, osteoclasts, as well as osteocytes) are themselves sensitive to mechanical loads and thus adapt to changing external conditions [5, 6],, it is now recognized that force application to the periodontal ligament (PDL) and the fibroblasts localized there is crucial for orthodontic tooth movement. However, the exact mode of force absorption and subsequent mechanotransduction have not been fully elucidated nor have all the molecular factors involved in this process been identified and functionally characterized. The number of signaling molecules associated with orthodontic tooth movement to date exceeds 100 and ranges from pro-inflammatory cytokines and "classical" coupling factors such as RANKL (Receptor Activator of NF-κB Ligand) and OPG (Osteoprotegerin) to autophagy regulators and hypoxia mediators [7]. What has actually gained increasing attention recently with regard to the regulation of bone remodeling during orthodontic tooth movement is the putative involvement of neuronal guidance molecules. This seems, at first sight, somewhat surprising as these molecules were originally identified for their functions in neuronal development, modulation of cell positioning and tissue and organ patterning [8], but a closer look shows that the importance of these molecules in the regulation of bone homeostasis and orthodontic tooth movement may still be underestimated. Indeed, NGMs mediated interactions between the nervous system and the skeleton have been shown in different contexts [9] and direct NGM effects on bone homeostasis [10] as well as their involvement in tooth development were shown [11]. Intriguingly, numerous NGMs are still expressed in adult dental tissue [12-14] and their function here is only beginning to be understood. Thus, the aim of this narrative review was to summarize known functions of NGMs in bone remodeling and bone homeostasis and to review the available literature on NGM functions in orthodontic tooth movement. We will therefore first briefly summarize the essential processes and the known molecular factors involved in physiological bone remodeling. Then, separately for each family, we will introduce those NGMs for which roles in bone remodeling have been described and explain their functions. Finally, we will discuss the NGMs previously associated with orthodontic tooth movement and explain their potential importance in this process.

Point 3:

For the same reason stated in my previous comment (this is a narrative review), how do the authors know that neutrins, ephrins and eph-receptors, semaphorins, and slits would be the only neuronal guidance molecules in bone remodeling to discuss about?

The same for neuronal guidance molecules in orthodontic tooth movement.

Response 3:

We very much understand the reviewer's concerns. In fact, we have included many sources in our narrative review based on our many years of experience with NGMs. However, in order to report the functions of NGMs for bone remodeling and, in particular, for orthodontic tooth movement as completely and unbiased as possible, we conducted a search in electronic databases in advance. The procedure is now presented in a newly created material and method section.

Revised text:

  1. Material and methods

This review is narrative in nature. Our own experience in the field of NGMs was a key factor in the selection of the studies cited. In order to reduce the possible bias, a supplementary literature search was conducted. Databases consulted in the project included: PubMed, EBSCO, Web of Science and Scopus. The following queries were used for the search: “(bone remodeling OR bone homeostasis OR bone) AND “(axon guidance molecules OR neuronal guidance molecules OR axon pathfinding OR axon guidance)”; (Orthodontic tooth movement AND (netrin* OR slit* OR semaphorin* OR ephrin* OR eph* OR eph-receptor*)”. Only studies in English language were selected. Editorial articles and commentaries were not excluded. We did not perform a specific quality assessment, which would hardly have been possible due to the basic research nature of most of the studies. However, we would like to apologize to authors whose work deserves to be mentioned, but which we have not quoted. this was not out of ignorance, but due to space restrictions.

Point 4:

There are some sentences in the text without reference to a previous study (or studies) in order to give evidence to their statements. Without references, these statements would be mere assumptions or allegations by the authors of the manuscript. Therefore, each of the following sentences need at least one reference to back up their statement: …

Response 4:

We would like to apologize for this confusion! We have added the required references in the introduction section. Concerning the references in the section on Ephrin/ Eph faction during tooth movement we thought that it might be sufficient to quote our own studies at the beginning of this section to which all statements made here refer to. However, this caused confusion. We apologize! According to the suggestion of the reviewer, we have linked the corresponding quotations again with the statements made and hope to have made the references clearer in this way.

Revised text:

What has actually gained increasing attention recently with regard to the regulation of bone remodeling during orthodontic tooth movement is the putative involvement of neuronal guidance molecules. This seems, at first sight, somewhat surprising as these molecules were originally identified for their functions in neuronal development, modulation of cell positioning and tissue and organ patterning [8], but a closer look shows that the importance of these molecules in the regulation of bone homeostasis and orthodontic tooth movement may still be underestimated. Indeed, NGMs mediated interactions between the nervous system and the skeleton have been shown in different contexts [9] and direct NGM effects on bone homeostasis [10] as well as their involvement in tooth development were shown [11]. Intriguingly, numerous NGMs are still expressed in adult dental tissue [12-14] and their function here is only beginning to be understood. Thus, the aim of this narrative review was to summarize known functions of NGMs in bone remodeling and bone homeostasis and to review the available literature on NGM functions in orthodontic tooth movement. Therefore, we will first briefly summarize the main processes and the known molecular factors involved in physiological bone remodeling. Then, separately for each family, we will introduce the NGMs for which a role in bone remodeling has been described and explain their functions. Finally, we will discuss the NGMs that have previously been associated with oral tooth movement and explain their potential importance in this process.

3.2.1. Ephrins and Eph-receptors

As mentioned above, functions in the maintenance of bone homeostasis have been defined for certain ephrin and Eph receptors. Within this context it should be noted once again that a distinctive feature of ephrin ligands is that, like their receptors, they are also capable of transmitting signals.

Bidirectional signaling between ephrin-B2 on osteoclasts and its receptor EphB4 on osteoblasts suppressed osteoclast differentiation while stimulating osteogenic differentia-tion of osteoblasts [47]. Likewise, bidirectional signaling, in this case between ephrin-A2 and EphA2 produced opposite results in osteoclasts and osteoblasts [49]. Taken together, bidirectional activation of ephrin-B2 and EphB4 resulted in bone formation whereas acti-vation of ephrin-A2 and EphA2 resulted in bone resorption. These data suggested a high importance of the ephrin-Eph system for bone homeostasis. Therefore, a possible role of these signaling pathways in bone remodeling during orthodontic tooth movement was investigated in our laboratory. As previously mentioned, fibroblasts of the PDL (PDLF) are the primary recipients of orthodontic forces. However, mechanically modulated expres-sion changes of ephrins had been described only in arterial endothelial cells [99,100]. Data for PDLF were not available at that time. In our studies, primary human PDLF responded to experimental mechanical stretch with an induction of ephrin-B2 expression and a de-crease in ephrin-A2 expression. Compression forces had an opposite effect [101,102]. Stretch induced ephrin-B2 expression via a signaling pathway involving FAK (Focal Ad-hesion Kinase), GTPase Ras, ERK1/2 (Extracellular signal Regulated Kinases) MAP (Mi-togen-Activated Protein) kinases, and transcription factor SP1 (Specifity Protein 1) [101]. Ephrin-B2-dependent osteogenic differentiation, clearly demonstrated by the expression of osteogenic differentiation markers and specific staining of mineralized matrix, in primary human osteoblasts of the alveolar ridge could be attributed to a signaling pathway acti-vated after receptor stimulation, which also involved Ras and ERK1/2 [101]. Compression forces caused induction of ephrin-A2 and attenuated ephrin-B2 in PDLF. In alveolar ridge osteoblasts, activation of the EphA2 receptor by exogenous ephirn-A2 resulted in suppres-sion of Ras activation and phosphorylation of ERK1/2, thereby omitting activation of os-teogenic differentiation markers such as RUNX2 or ALPL [103]. Different mechanical stimulations therefore led to selective induction of either ephrin-B2 or ephrin-A2, each with reciprocal consequences for osteoblasts of the alveolar ridge. Unlike the induction of ephrin-B2 after stretch, initially it remained unclear how ephrin-A2 is induced after me-chanical stimulation with compression forces in PDLF. Using three independent PDLF populations, we showed that the induction of ephrin-A2 mRNA after compression is ac-companied by, compared with activation after stretch, delayed phosphorylation and transcriptional activation of c-fos, which heterodimerizes with c-jun to the AP-1 (activator protein-1) transcription factor. Compression-dependent induction of c-fos and ephrin-A2 was also demonstrated at the protein level. To prove the involvement of both ERK1/2 and c-fos, both factors were selectively inhibited. Activation of ERK1/2 was prevented by the use of U0126 a specific inhibitor of ERK1/2 kinase MEK1 (MAP2K1), and inactivation of c-fos at the transcriptional level was achieved by using si (small-interference) RNA. Both strategies resulted in a significant decrease in ephrin-A2 expression, thus demonstrating the direct involvement of both GTPase Ras, ERK1/2, and c-fos in the compres-sion-dependent induction of ephrin-A2 in PDLF (Figure 1) [102]. Osteogenic differentia-tion of subpopulations of PDLFs contribute to bone formation during tooth movement, thus loss of this potential is relevant to bone remodeling during tooth movement. In pre-vious experiments, we observed a marked ephrin-A2-dependent decrease in the expres-sion of osteogenic markers in both alveolar ridge osteoblasts and PDLFs. Interestingly, this was even more pronounced in PDLF than in osteoblasts. A possible cause of this enhanc-ing effect could be a higher abundance of ephrin-A2 in the PDLF population. Indeed, PDLF showed a significant induction of ephrin-A2 after EphA2 activation by exogenous ephrin-A2. This was accompanied by rapid activation of ERK1/2 and induction of c-fos. This implies that ephrin-A2 may enhance its own expression in the PDLF population, thereby contributing to the marked effect of ephrin-A2 on the inhibition of osteogenic dif-ferentiation in PDLF [102].

Our in vitro studies [101-103] have recently been complemented by animal studies by other authors that also provided evidence for the involvement of the Ephrin-Eph pathway in the regulation of bone remodeling during orthodontic tooth movement. Hou and col-leagues used a rat model of orthodontic tooth movement and demonstrated strong ephrin-B2 staining in compressed periodontal tissues including PDLF and osteoclasts. More insight into ephrin-B2 dependent osteoclast activation was gained using RAW264.7 murine macrophage cells were compression induced the expression of ephrin-B2 along with the osteoclast markers nuclear factor of activated T cells cytoplasmic 1 (NFATc1) and calcitonin receptor (CTR). Moreover, the authors confirmed compression force dependent decreases of ephrin-B2 and EphB4 in ST2 murine stromal cells which are able to differen-tiate into osteoblast-like cells in response to osteogenic stimuli in vitro [104]. Only very re-cently Jiang et al. focused on investigating an ephrin-Eph role on the tension side of or-thodontic tooth movement. Using also a rat model it was shown that local administration of a selective small molecule EphB4 receptor inhibitor targeting the kinase domain, shifted bone remodeling activity toward bone resorption after application of orthodontic forces, because EphB4 forward signaling was blocked in osteoblasts resulting in attenuated os-teoblastic differentiation, while the extracellular domain of the EphB4 receptor was still able to initiate Ephrin-B2 reverse signaling and, thus, osteoclastogenesis [105].

Point 5:

The authors might want to give a number to the following references in the text:

“Tooth development is a complex process that requires the mutual interaction of embryonic oral epithelial cells and ectomesenchymal cells from the neural crest (Jernvall and Thesleff, 2000).”

“To further investigate the function of ephrin-A2 in osteoclasts, Irie and colleagues (…)”

“EphA8 in the trigeminal ganglion plays a major role in tooth development (Luukko et al., 2005).”

“In primary dental pulp stem cells ephrin-B1-EphB2 interaction induced osteogenic and odontoblastogenic differentiation (Wang et al., 2013; Wang et al., 2014).”

“However, Xu and colleagues and Li and colleagues reported conflicting data”

Response 5:

We would like to apologize for these embarrassing errors and would like to blame the majority of them on a malfunction of our reference manager. We have added the corresponding references in the specific sections and in the reference section.

Revised text:

Tooth development is a complex process that requires the mutual interaction of embryonic oral epithelial cells and ectomesenchymal cells from the neural crest [37]. Netrin 1 and Netrin-3 are expressed in the developing tooth germ and might be involved in axon growth during tooth development. Netrin-1 deficient mice, however, did not show axon growth or tooth innervation alterations.

To further investigate the function of ephrin-A2 in osteoclasts, Irie and colleagues [49] transduced osteoclast progenitor cells with a retroviral vector encoding ephrin-A2.

In particular, the expression of the ephrins and Eph receptors ephrin-A1-A5 and EphA2, EphA3, EphA4, EphA7, and EphA8 in the trigeminal ganglion plays a major role in tooth development [14].

The importance of ephrins and Eph receptors in tooth development is also underscored by in vitro data: In primary dental pulp stem cells ephrin-B1-EphB2 interaction induced osteogenic and odontoblastogenic differentiation [64].

However, Xu and colleagues [98]reported conflicting data: osteoblast-specific Slit3-deficient mice showed a decrease in bone mass and osteoclast-specific Slit3-deficient mice had no change in bone mass. In their model Slit3 was strongly expressed in osteoblasts but hardly expressed in osteoclasts; a finding which was confirmed by the work of Li and colleagues [95]. Thus, the actual role of Slit3 in bone remodeling cannot be defined conclusively as yet.

Round 2

Reviewer 2 Report

The manuscript now seems to be suitable for publication.

Author Response

Manuscript ijms- 1869450 “Neuronal guidance molecules in bone remodeling and orthodontic tooth movement”

We thank the editor and reviewers for the close reading of our manuscript, for the interest in our work and for the valuable suggestions for improvement. We have revise our manuscript accordingly.

Response to the Editor

Point 1:

  1. Since a substantial part of the review is on NGM and bone remodelling, we suggest to revise the title (e.g., Neuronal guidance molecules in bone remodelling and orthodontic tooth movement, or the like) so that both aspects are covered.

Response 1:

We are happy to follow the editor's welcome suggestion and change the title according to the suggestion. At the same time we thank you for this decisive improvement of the title.

Revised text:

Neuronal guidance molecules in bone remodeling and ortho-dontic tooth movement

Point 2:
2. In response to a reviewer comment (“....how do the authors know that netrins, ephrins and eph-receptors, semaphorins, and slits would be the only neuronal guidance molecules in bone remodeling to discuss about?) you added a Material and Methods section. Since this does not fit well with the format of a review, we suggest to either integrate a shortened version of the process into the text or move the section to the supplement. Also, from this text, the choice of the 4 groups of molecules still does not get clear. As you stated in your letter, the choice is mainly based on your research interest. While this is certainly an acceptable explanation, I would still like to suggest that you at least name those molecule groups that are not included and/or mention a more general review on NGM family members.

Response 2:

We are grateful for the editor's suggestion. We have therefore removed the material and methods section and appended it to the introduction in an abbreviated form.

We have also tried to make it clear in the abstract and in the methods section that we have investigated the possible involvement of all four known NGM families. We hope this is now clear in the present version of the manuscript.

Revised text:

During orthodontic tooth movement, mechanically induced remodeling occurs in the alveolar bone due to the action of orthodontic forces. The number of factors identified to be involved in mechanically induced bone remodeling is growing steadily. With the uncovering of functions of neuronal guidance molecules (NGMs) for skeletal development as well as for bone homeostasis, NGMs are now also among the potentially significant factors for the regulation of bone remodeling during orthodontic tooth movement. This narrative review attempts to summarize the functions of NGMs in bone homeostasis and provides insight into the, presently scarce, literature on the functions of these molecules during orthodontic tooth movement. Presently four families of NGMs are known: Netrins, Slits, Semaphorins, ephrins and Eph receptors. Search of electronic databases revealed roles in bone homeostasis of representatives from all four NGM families. Functions during orthodontic tooth movement, however, were only identified for Semaphorins, ephrins and Eph receptors. For these crucial prerequisites for participation in the regulation of orthodontically induced bone remodeling, such as expression in cells of the periodontal ligament and in the alveolar bone, as well as mechanical inducibility, were shown which suggests that the importance of NGMs in orthodontic tooth movement may be underappreciated to date and further research might be warranted.

We should like to point out that this review is narrative in nature. In order to reduce the possible bias, a literature search including all four known families of NGMs was conducted. Databases consulted in the project included: PubMed, EBSCO, Web of Science and Scopus. The following queries were used for the search: “(bone remodeling OR bone homeostasis OR bone) AND “(axon guidance molecules OR neuronal guidance molecules OR axon pathfinding OR axon guidance)”; (Orthodontic tooth movement AND (netrin* OR slit* OR semaphorin* OR ephrin* OR eph* OR eph-receptor*)”. We did not perform a specific quality assessment, which would hardly have been possible due to the basic research nature of most of the studies.

Point 3:

The part on Netrins is missing a quite early publication on Netrin-1 from Mediero et al. (2015) on the effect of Netrin-1 deficiency for osteoclastogenesis. I would suggest to include it.

Response 3:

Despite the best of intentions to consider highly relevant studies, we can be blamed for an omission here. We thank you for your attention and have now included the relevant reference and briefly described its content.

Revised text:

During osteoclast differentiation, osteoclast precursors increase expression of Netrin-1 and its receptor Unc5b. Netrin-1 binding to Unc5b promoted osteoclast differentiation in vitro, which could be prevented by antibody-mediated blockade of Netrin-1 or Unc5b. These in vitro findings were confirmed in Netrin-1-deficient(Ntn1-/-) mice [33].

  1. Mediero, A.; Ramkhelawon, B.; Perez-Aso, M.; Moore, K. J.; Cronstein, B. N., Netrin-1 is a critical autocrine/paracrine factor for osteoclast differentiation. J Bone Miner Res 2015, 30, (5), 837-54.

Point 4:

For the conclusion, we would appreciate if you could take into the account the topic of the special issue (From Molecular Pathways to Translational Research) and add the translational aspects of this group of molecules.

Response 4:

We would like to apologize for this omission and have now tried to bring in a translational aspect in the conclusion section.

Revised text:

… However, their potential role should not be underestimated, as they possess a number of properties that predestine them as regulatory molecules during orthodontic tooth movement: many NGMs are involved in bone development and homeostasis, some have been shown to be directly regulated by key factors of bone homeostasis, their functions include the control of innervation and angiogenesis, both necessary during bone remodeling many are expressed in adult hard tissues, some have been shown to be mechanosensitive in different cell types, all this suggests that the importance of NGMs in bone remodeling and thus orthodontic tooth movement may be underappreciated to date and further research is therefore absolutely warranted. As the further elucidation of the mechanisms by which NGMs are involved in bone remodeling in general and in orthodontic tooth movement in particular could pave the way for translational development of new therapeutics for the treatment of skeletal diseases as well as open up possibilities to perform orthodontic tooth movement without side effects.
